# Adversarial Attacks on Medical Hyperspectral Imaging Exploiting Spectral-Spatial Dependencies and Multiscale Features

## Abstract

Medical hyperspectral imaging (HSI) represents a transformative innovation in diagnosing diseases and planning treatments by capturing detailed spectral and spatial features of tissues. However, the integration of deep learning into medical HSI classification has unveiled critical vulnerabilities to adversarial attacks. These attacks compromise the reliability of clinical applications, potentially leading to diagnostic inaccuracies and jeopardizing patient outcomes. This study identifies two fundamental reasons for the susceptibility of medical HSI models to adversarial manipulation: their reliance on local pixel dependencies, which are essential for preserving tissue structures, and their dependence on Multiscale spectral-spatial features, which encode hierarchical tissue information. To address these vulnerabilities, we propose a novel adversarial attack framework specifically tailored to medical HSI. Our approach introduces the Local Pixel Dependency Attack, which exploits spatial relationships between neighboring pixels, and the Multiscale Information Attack, which perturbs spectral and spatial features across hierarchical scales. Experiments on the Brain and MDC datasets reveal that our method significantly reduces classification accuracy, particularly for critical tumor regions, while maintaining imperceptible perturbations. Compared to existing methods, the proposed framework highlights the unique fragility of medical HSI models and underscores the urgent need for robust defenses. This work highlights critical vulnerabilities in medical HSI models and demonstrates how leveraging local pixel dependencies and Multiscale spectral-spatial features can guide the development of targeted defenses to enhance model robustness and clinical reliability.

## 1 Introduction

Hyperspectral imaging (HSI) is a powerful technique that captures a wide spectrum of light across contiguous bands, enabling detailed analysis of material and surface characteristics. Originally developed for agriculture, environmental monitoring, and land cover classification, HSI is increasingly applied in medical imaging due to its ability to detect subtle biochemical and structural variations beyond the capabilities of conventional modalitiesLu & Fei (2014). By integrating spatial and spectral information, HSI supports precise tissue analysis for tasks such as tumor detection, vascular visualization, and histopathological segmentationLu et al. (2014); Channing (2022). The combination with advanced computational methods has further improved feature extraction and diagnostic performanceCui et al. (2022); Xiang et al. (2023).

Despite these advances, the integration of HSI with deep learning introduces vulnerability to adversarial attacks—small, imperceptible perturbations that can severely mislead models Kumar et al. (2024); Shen et al. (2025). This is especially concerning in medical applications, where misclassification may compromise patient safety Goodfellow et al. (2014). Studies on diabetic retinopathy grading and Alzheimer's disease prediction confirm that minor perturbations can cause significant performance drops Cheng et al. (2024); Baytaş (2024). HSI models, which rely on rich spectral-spatial features, are particularly susceptible to such attacks Zeng et al. (2023); Mangotra et al. (2023).

In hyperspectral remote sensing, CNN-based models dominate classification tasks due to their ability to process high-dimensional data and learn complex spectral-spatial features Khan et al. (2018). Ex-

isting adversarial attacks mainly focus on pixel-level perturbations, assuming classification depends only on spectral information from individual pixels. However, in hyperspectral medical images, spatial dependencies between neighboring pixels are critical Khan et al. (2021). Tissue structure and tumor boundaries are often defined by local spatial patterns, and minor local variations can significantly affect classification Xie et al. (2023). As a result, current adversarial methods often underperform in medical HSI (MHSI), where such spatial context is indispensable.

Furthermore, compared with remote sensing, MHSI contains more complex and Multiscale information. Medical images are smaller in spatial scale but demand higher resolution to distinguish subtle variations in tissue, vasculature, and abnormalities Fei (2019). Fine-grained tumor characteristics require local-scale analysis, while broader structures such as vascular networks benefit from global-scale interpretation Lu & Fei (2014). Unlike remote sensing imagery focused on large-scale patterns Peng et al. (2024), MHSI involves intricate Multiscale features essential for accurate diagnosis Wei et al. (2019); Zeng et al. (2023).

Despite progress, most adversarial methods overlook these characteristics, focusing solely on spectral or pixel-level perturbations Shi et al. (2022). This gap limits their effectiveness in medical contexts and calls for more tailored attack strategies that exploit spatial dependencies and Multiscale information.

In this paper, our main contributions are as follows:

- We identify the unique vulnerabilities of MHSI models stemming from local pixel dependencies and Multiscale spectral-spatial structures.

- We propose two novel attack methods: **Local Pixel Dependency Attack** and **Multiscale Information Attack**, designed to exploit these characteristics.

- We demonstrate through experiments that our methods substantially reduce classification accuracy in critical regions while preserving perturbation imperceptibility, highlighting the urgent need for dedicated defense mechanisms.

## 2 RELATED WORKS

### 2.1 HYPERSPECTRAL IMAGE CLASSIFICATION

Hyperspectral image classification has progressed from traditional methods (PCA, SVM, KNN) to deep learning approaches that automate feature extraction and integrate spectral-spatial informationLi et al. (2019). While computationally efficient, traditional methods require extensive feature engineering for high-dimensional dataKumar et al. (2020). Convolutional Neural Networks (CNNs) significantly advanced the field by exploiting spectral-spatial correlations. Early 2D-CNNs extracted spatial features, while 3D-CNNs enabled spectral-spatial integration. Hybrid architectures like HybridSN balance efficiency and accuracy through combined 3D-2D convolutionsRoy et al. (2019). The Spectral-Spatial Residual Network(SSRN) further improved generalization via residual learningZhong et al. (2017). The Self-attention Context Network(SACNet) employs self-attention and context encoding to capture global dependencies, improving robustness through hierarchical feature extractionXu et al. (2021). More recently, graph-based methods with uncertainty quantification have been explored to improve reliability in OOD and misclassification detectionYu et al. (2024).

### 2.2 MEDICAL HYPERSPECTRAL IMAGE CLASSIFICATION

Medical HSI (MHSI) classification provides critical diagnostic insights but faces high-dimensionality challenges. CNN-based methods effectively capture hierarchical spectral-spatial features and have been widely applied to tumor and lesion analysisHuang et al. (2019). Recent works emphasize the importance of pixel dependencies and multiscale structures, demonstrating that accurate modeling of local spatial coherence and hierarchical spectral-spatial patterns is crucial for robust medical diagnosis under perturbationsXie et al. (2023); Wei et al. (2019). More recently, the Dual-Stream model has been introduced for medical hyperspectral classification, integrating complementary spatial and spectral streams to enhance discriminative powerYun et al. (2023).

### 2.3 ADVERSARIAL ATTACK AND DEFENSE ON HYPERSPECTRAL IMAGES

Adversarial attacks exploit imperceptible perturbations to induce misclassification. For hyperspectral data, several representative attack methods have been developed. Spectral–Spatial FGSM(SS-FGSM) perturbs spectral features at the pixel levelShi et al. (2023), while Spectral–Spatial Attack(SSA) explores spectral-spatial adversarial strategiesYin et al. (2025). Multifeature collaborative adversarial Network(MfcaNet) introduces multi-feature collaborative perturbations to enhance attack success ratesShi et al. (2022). These approaches, however, were not specifically designed for medical HSI, leaving vulnerabilities in clinical applications.

In parallel, defense networks have been proposed to counter such threats. The Robust Class Context-Aware(RCCA) network leverages contextual class dependencies to enhance robustness against adversarial perturbationsTu et al. (2023). The Weighted Fusion of Spectral Transformer and Spatial Self-Attention(WFSS) integrates transformer-based spectral modeling with spatial self-attention for multi-level defenseTang et al. (2024). More recently, attention-based defenses such as Attack-Invariant Attention Feature(AIAF) and Spatial-spectral self-attention Network($S^3$ANet) have been introduced, focusing on adaptive information aggregation and spectral–spatial alignment to suppress adversarial noise while preserving lesion structuresShi et al. (2024); Xu et al. (2024). These methods represent strong defensive baselines, and we include them in our experiments to comprehensively evaluate the effectiveness of our proposed attack framework.

## 3 METHOD

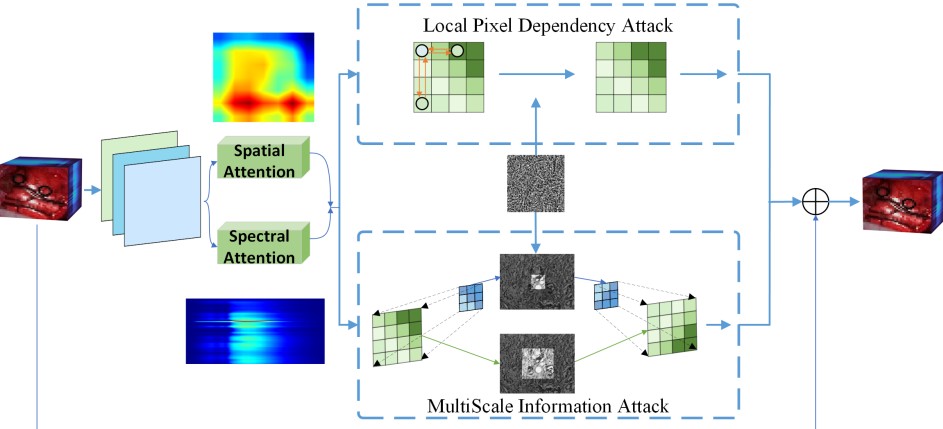

Figure 1: The proposed adversarial attack and defense framework for HSI classification.

### 3.1 LOCAL PIXEL DEPENDENCY ATTACK

In medical hyperspectral imaging, adversarial attacks present significant challenges due to the intricate spatial relationships between neighboring pixels. To address this issue, we introduce the Local Pixel Dependency Attack, which leverages these local dependencies between adjacent pixels to generate adversarial examples. The goal of this attack is to mislead the model by introducing perturbations that are perceptually subtle yet highly effective in altering the classification outcome.

The core idea behind the Local Pixel Dependency Attack is to exploit the spatial relationships between neighboring pixels. By incorporating these local pixel dependencies, we apply gradient-based perturbations in a manner that ensures the attack remains both effective and minimally detectable.

We formulate the attack as an optimization problem, where the objective is to perturb the input hyperspectral image, denoted as $\mathbf{x}$, such that the model misclassifies it, while accounting for the local pixel dependencies. Specifically, given an image $\mathbf{x}$ and its true label $y$, the goal is to minimize the loss function with respect to a target label $\hat{y}$, while introducing perturbations that preserve the spatial coherence between adjacent pixels. The gradient of the loss function $\mathscr{L}$ with respect to the

input image $\mathbf{x}$ is calculated as:

$$\nabla_{\mathbf{x}}\mathscr{L}(\mathbf{x},y) \tag{1}$$

The objective of the attack can be expressed as:

$$\mathbf{x}_{\text{adv}} = \mathbf{x} - \varepsilon \cdot \frac{1}{N_{i,j}} \sum_{(i',j')\in\mathscr{W}(i,j)} \nabla_{\mathbf{x}}\mathscr{L}(\mathbf{x},y), \tag{2}$$

where $\mathbf{x}$ represents the original hyperspectral image, while $\mathbf{x}_{\text{adv}}$ is the adversarial image obtained after applying perturbations. The perturbation strength is controlled by the parameter $\varepsilon$, which determines how much the image is altered in each iteration. The term $\nabla_{\mathbf{x}}\mathscr{L}(\mathbf{x},y)$ refers to the gradient of the loss function $\mathscr{L}$ with respect to the image $\mathbf{x}$, which is used to guide the perturbation direction. The sum is taken over a local window $\mathscr{W}(i,j)$, which includes neighboring pixels of a given pixel $(i,j)$. The number of pixels in this local window is denoted by $N_{i,j}$, and it ensures that the perturbation considers the pixel dependencies within the local region.

The gradient is averaged over the local window for each pixel, ensuring that perturbations account for the relationships between adjacent pixels. This averaging step is important for maintaining the spatial structure of the image and making the attack more imperceptible.

The attack is applied iteratively to refine the adversarial perturbations. In each iteration, the image is updated as follows:

$$\mathbf{x}^{(t+1)} = \mathbf{x}^{(t)} - \varepsilon \cdot \frac{1}{N_{i,j}} \sum_{(i',j')\in\mathscr{W}(i,j)} \nabla_{\mathbf{x}}\mathscr{L}(\mathbf{x},y), \tag{3}$$

where $\mathbf{x}^{(t)}$ is the image at the $t$-th iteration and $\mathbf{x}^{(t+1)}$ is the updated image after perturbation. This process continues for a predefined number of iterations, typically 20 in our case, to ensure that the perturbations gradually mislead the model while preserving local pixel dependencies.

In the targeted attack scenario, the objective is to misclassify the hyperspectral image $\mathbf{x}$ into a specific target label $\hat{y}$, as opposed to just any incorrect label. The loss function is modified to encourage the model to predict the target label for the perturbed image:

$$\mathscr{L}(\mathbf{x}_{\text{adv}},\hat{y}) = -\log P(\hat{y}|\mathbf{x}_{\text{adv}}) \tag{4}$$

### 3.2 MULTISCALE INFORMATION ATTACK

In medical hyperspectral imaging, adversarial attacks must account for the Multiscale nature of spectral and spatial features. To address this, we propose the Multiscale Information Attack, which generates perturbations by leveraging both downsampling and upsampling operations at multiple scales. This approach targets the Multiscale dependencies inherent in hyperspectral data, crafting perturbations that disrupt model predictions across different resolutions.

The Multiscale Information Attack processes the input hyperspectral image $\mathbf{x} \in \mathbb{R}^{B\times D\times H\times W}$, where $B$ is the batch size, $D$ is the number of spectral bands, and $H,W$ are the spatial dimensions. For each scale factor $s \in \mathscr{S}$, the attack introduces perturbations in the scale space. Specifically, each spectral band $\mathbf{x}^{(d)}$ is first downsampled to a lower resolution:

$$\mathbf{x}_{\text{down}}^{(d,s)} = \text{Downsample}(\mathbf{x}^{(d)},s) \tag{5}$$

At the downsampled resolution, perturbations are introduced as follows:

$$\mathbf{x}_{\text{pert}}^{(d,s)} = \mathbf{x}_{\text{down}}^{(d,s)} + \varepsilon \cdot \nabla_{\mathbf{x}_{\text{down}}^{(d,s)}}\mathscr{L}(\mathbf{x},y), \tag{6}$$

where $\mathscr{L}(\mathbf{x},y)$ is the loss function guiding the attack, and $\varepsilon$ controls the strength of the perturbation. After adding the perturbation, the image is upsampled back to the original resolution:

$$\mathbf{x}_{\text{up}}^{(d,s)} = \text{Upsample}(\mathbf{x}_{\text{pert}}^{(d,s)},(H,W)) \tag{7}$$

For each scale $s$, the perturbations across all spectral bands are summed to produce the total perturbation at that scale:

$$\mathbf{p}_s = \sum_{d=1}^{D} \mathbf{x}_{\text{up}}^{(d,s)} \tag{8}$$

To incorporate Multiscale information, the perturbations from all scales are aggregated:

$$\mathbf{p} = \sum_{s \in \mathscr{S}} \mathbf{p}_s \tag{9}$$

Finally, the adversarial example is generated by adding the aggregated perturbation to the original input image:

$$\mathbf{x}_{\text{adv}} = \mathbf{x} + \varepsilon \cdot \mathbf{p} \tag{10}$$

By introducing perturbations at multiple resolutions and restoring them to the original size, the Multiscale Information Attack ensures that the adversarial perturbations impact both fine-grained and coarse-grained spatial features in the hyperspectral data. This hierarchical perturbation strategy aligns with the Multiscale nature of hyperspectral image analysis, making it particularly effective in medical applications.

### 3.3 ADVERSARIAL ATTACK FRAMEWORK

The final adversarial perturbation is the combination of the local pixel-dependent perturbation and the Multiscale perturbation:

$$\delta_{\text{final}} = \delta_{\text{local}} + \delta_{\text{Multiscale}} \tag{11}$$

The adversarial example is then generated as:

$$x_{\text{adv}} = x + \delta_{\text{final}}, \tag{12}$$

where $x$ is the original hyperspectral image, and $x_{\text{adv}}$ is the perturbed adversarial example.

### 3.4 ALGORITHM SUMMARY

The general process of the proposed attack method can be briefly summarized as Algorithm 1:

---

**Algorithm 1** Overall Framework for Adversarial Attacks in MHSI

---

**Require:** HSI data $\mathbf{x}$, target $\hat{y}$, model $f$, $\varepsilon$, iterations $T$, scales $\mathscr{S}$
**Ensure:** Adversarial $\mathbf{x}_{\text{adv}}$
 1: $\mathbf{x}_{\text{adv}} \leftarrow \mathbf{x}$
 2: **for** $t = 1$ **to** $T$ **do**
 3:     **Local Pixel Attack:**
 4:     **for** each pixel $(i, j)$ in $\mathbf{x}_{\text{adv}}$ **do**
 5:         $\mathscr{W}(i, j) \leftarrow$ local window; $N_{i,j} \leftarrow |\mathscr{W}|$
 6:         $\nabla_{\mathbf{x}} \mathscr{L} \leftarrow$ loss gradient of $\mathbf{x}_{\text{adv}}$
 7:         $\bar{\nabla}_{\mathbf{x}} \leftarrow \frac{1}{N_{i,j}} \sum_{\mathscr{W}(i,j)} \nabla_{\mathbf{x}} \mathscr{L}$
 8:         $\mathbf{x}_{\text{adv}}(i, j) \leftarrow \mathbf{x}_{\text{adv}}(i, j) - \varepsilon \cdot \bar{\nabla}_{\mathbf{x}}$
 9:     **end for**
10:     **Multiscale Attack:**
11:     $\mathbf{p} \leftarrow \mathbf{0}$
12:     **for** $s \in \mathscr{S}$ **do**
13:         $\mathbf{p}_s \leftarrow \mathbf{0}$
14:         **for** band $d$ **do**
15:             $\mathbf{x}_{\text{down}} \leftarrow \text{Downsample}(\mathbf{x}^{(d)}, s)$
16:             $\mathbf{x}_{\text{pert}} \leftarrow \mathbf{x}_{\text{down}} + \varepsilon \cdot \nabla_{\mathbf{x}_{\text{down}}} \mathscr{L}(\mathbf{x}_{\text{down}}, \hat{y})$
17:             $\mathbf{x}_{\text{up}} \leftarrow \text{Upsample}(\mathbf{x}_{\text{pert}}, (H, W))$
18:             $\mathbf{p}_s \leftarrow \mathbf{p}_s + \mathbf{x}_{\text{up}}$
19:         **end for**
20:         $\mathbf{p} \leftarrow \mathbf{p} + \mathbf{p}_s$
21:     **end for**
22:     $\mathbf{x}_{\text{adv}} \leftarrow \mathbf{x}_{\text{adv}} + \varepsilon \cdot \mathbf{p}$
23: **end for**
24: **return** $\mathbf{x}_{\text{adv}}$

---

## 4 EXPERIMENTS

### 4.1 DATASETS

#### 4.1.1 IN-VIVO HYPERSPECTRAL HUMAN BRAIN IMAGE DATABASE FOR BRAIN CANCER DETECTION

The In-Vivo Hyperspectral Human Brain Image Database for Brain Cancer Detection consists of 36 hyperspectral images collected from 22 neurosurgical operationsFabelo et al. (2019). It covers four annotated classes: normal tissue, tumor tissue, blood vessels, and background elements. The images span the Visual and Near-Infrared (VNIR) spectrum from 400 to 1000 nm, providing over 300,000 labeled spectral signatures. Labels were generated using a semi-automatic methodology based on the Spectral Angle Mapper (SAM) algorithm, cross-referenced with histopathological evaluations. This dataset serves as a significant resource for developing machine learning models for brain tumor classification and guiding real-time surgical decisions.

#### 4.1.2 MULTIDIMENSIONAL CHOLEDOCH (MDC) DATASET

MultiDimensional Choledoch (MDC) Dataset includes 880 hyperspectral scenes collected from 174 individuals, comprising 689 scenes with partial cancer regions (L), 49 with complete cancerous areas (N), and 142 without cancer (P)Zhang et al. (2019). This dataset only uses binary classification to determine the cancer region from the normal region. The hyperspectral data were captured using a system with a 20× objective lens, covering wavelengths from 450 nm to 1000 nm with 60 spectral bands per scene. Each hyperspectral image was resized to 256×320 pixels to enhance computational efficiency.

### 4.2 EXPERIMENTAL SETUP

In this study, we performed experiments on hyperspectral image datasets, specifically targeting medical image classification tasks. To reduce the data dimensionality and extract the most important spectral features, Principal Component Analysis (PCA) was applied, reducing the spectral dimensions to 20 components. This reduction in dimensionality helps minimize computational overhead while retaining the essential spectral information for classification.

For data preprocessing, image cubes were generated using a sliding window approach with a window size of 11×11, allowing for the extraction of local spatial-spectral features. Zero-padding was applied at the borders to handle edge pixels, which do not have enough neighboring pixels for patch extraction. We used a training-to-testing split of 80% for training and 20% for testing, ensuring that the training set included a diverse representation of different classes.

### 4.3 EVALUATION METRICS

Since we are conducting medical image adversarial attacks, attacking the lesion area to misclassify it into normal areas will cause the greatest harm to patients and the medical system. Therefore, our evaluation metrics mainly focus on the classification success rate of the lesion areas in each dataset. The lower the success rate, the better the effectiveness of our attack. At the same time, we also adopt three commonly used metrics for comprehensive evaluation:

**Overall Accuracy (OA)** measures the overall proportion of correctly classified pixels. It is defined as:

$$\text{OA} = \frac{\sum_{i=1}^{C} N_{ii}}{\sum_{i=1}^{C} \sum_{j=1}^{C} N_{ij}} \tag{13}$$

where $N_{ij}$ represents the number of pixels whose ground truth class is $i$ and predicted class is $j$, and $C$ is the total number of classes.

**Average Accuracy (AA)** calculates the mean classification accuracy across all classes, reflecting the model's balanced performance:

$$\text{AA} = \frac{1}{C} \sum_{i=1}^{C} \frac{N_{ii}}{\sum_{j=1}^{C} N_{ij}} \tag{14}$$

| target model | attack method | Normal Tissue(↓) | Tumor Tissue(↑) | Hyper vascularized(↓) | Background (↓) | OA | AA | KAPPA | L0 | L2 |
|---|---|---|---|---|---|---|---|---|---|---|
| HybridSNRoy et al. (2019) | MfcaNetShi et al. (2022) | 0 | 85.81 | 0 | 0 | 96.61 | 78.45 | 95.02 | 3428 | 12.2 |
| | SSAYin et al. (2025) | 0 | 86.53 | 0 | 0 | 96.61 | 78.42 | 95.01 | 4687 | 13.6 |
| | SS-FGSMShi et al. (2023) | 0 | 82.16 | 0 | 0 | 96.61 | 78.44 | 95.02 | 3274 | 11.3 |
| | Ours | 0 | 92.02 | 0 | 0 | 96.59 | 78.3 | 94.98 | 1896 | 7.8 |
| SSRNZhong et al. (2017) | MfcaNetShi et al. (2022) | 0 | 85.25 | 0 | 0 | 96.61 | 78.47 | 95.03 | 3512 | 12.5 |
| | SSAYin et al. (2025) | 0 | 78.14 | 0 | 0 | 96.63 | 78.68 | 95.08 | 4825 | 14.1 |
| | SS-FGSMShi et al. (2023) | 0 | 88.06 | 0 | 0 | 96.6 | 78.32 | 95.03 | 3341 | 11.5 |
| | Ours | 0 | 95.35 | 0 | 0 | 96.58 | 78.9 | 94.95 | 1958 | 8.1 |
| SacNetXu et al. (2021) | MfcaNetShi et al. (2022) | 0 | 81.72 | 0 | 0 | 96.62 | 78.57 | 95.05 | 3395 | 11.9 |
| | SSAYin et al. (2025) | 0 | 74.33 | 0 | 0 | 96.64 | 78.82 | 95.12 | 4973 | 14.7 |
| | SS-FGSMShi et al. (2023) | 0 | 83.46 | 0 | 0 | 96.62 | 78.51 | 95.04 | 3189 | 11 |
| | Ours | 0 | 91.54 | 0 | 0 | 96.59 | 78.32 | 94.99 | 2027 | 8.5 |
| UAGCNYu et al. (2024) | MfcaNetShi et al. (2022) | 0 | 84.92 | 0 | 0 | 96.68 | 78.61 | 95.07 | 3315 | 11.8 |
| | SSAYin et al. (2025) | 0 | 79.85 | 0 | 0 | 96.71 | 78.96 | 95.15 | 4728 | 14.3 |
| | SS-FGSMShi et al. (2023) | 0 | 85.74 | 0 | 0 | 96.66 | 78.55 | 95.02 | 3092 | 10.9 |
| | Ours | 0 | 93.28 | 0 | 0 | 96.62 | 78.41 | 94.97 | 1975 | 7.9 |
| Dual-StreamYun et al. (2023) | MfcaNetShi et al. (2022) | 0 | 71.41 | 0 | 0 | 96.68 | 82.15 | 94.22 | 3607 | 13 |
| | SSAYin et al. (2025)citehynu2023patent | 0 | 67.66 | 0 | 0 | 95.8 | 83.09 | 93.75 | 4896 | 14.4 |
| | SS-FGSMShi et al. (2023) | 0 | 62.55 | 0 | 0 | 95.96 | 84.36 | 94.46 | 3425 | 11.8 |
| | Ours | 0 | 80.75 | 0 | 0 | 95.48 | 79.81 | 92.48 | 2144 | 8.7 |
| RCCATu et al. (2023) | MfcaNetShi et al. (2022) | 0.45 | 16.39 | 0.17 | 0.06 | 99.09 | 95.54 | 98.55 | 2968 | 10.4 |
| | SSAYin et al. (2025) | 1.73 | 24.18 | 0.39 | 0.63 | 98.31 | 93.27 | 96.91 | 4087 | 11.9 |
| | SS-FGSMShi et al. (2023) | 0.42 | 20.86 | 0.11 | 0.45 | 98.72 | 94.68 | 97.08 | 2854 | 10.9 |
| | Ours | 1.22 | 31.41 | 0.69 | 1.25 | 97.94 | 92.95 | 95.28 | 1718 | 7 |
| WFSSTang et al. (2024) | MfcaNetShi et al. (2022) | 1.17 | 28.02 | 0.58 | 0.42 | 98.07 | 93.45 | 98.04 | 2897 | 10.7 |
| | SSAYin et al. (2025) | 0.88 | 17.96 | 0.03 | 0.09 | 98.93 | 95.29 | 98.97 | 4149 | 12.2 |
| | SS-FGSMShi et al. (2023) | 1.28 | 32.05 | 0.85 | 1.15 | 97.86 | 93.16 | 95.13 | 2766 | 10.8 |
| | Ours | 1.28 | 37.68 | 1.37 | 1.63 | 96.91 | 90.76 | 93.57 | 1684 | 6.8 |
| AIAFShi et al. (2024) | MfcaNetShi et al. (2022) | 0.3 | 14.8 | 0.04 | 0.08 | 98.72 | 96.22 | 98.25 | 3021 | 10.9 |
| | SSAYin et al. (2025) | 1.1 | 18.55 | 0.37 | 0.29 | 98.39 | 94.94 | 97.62 | 4012 | 11.7 |
| | SS-FGSMShi et al. (2023) | 0.38 | 16.88 | 0.2 | 0.12 | 98.63 | 95.64 | 98.02 | 2879 | 11 |
| | Ours | 0.85 | 29.66 | 1.03 | 0.79 | 98.01 | 91.87 | 96.62 | 1741 | 7.1 |
| S$^3$ANetXu et al. (2024) | MfcaNetShi et al. (2022) | 0.26 | 13.98 | 0.09 | 0.02 | 98.77 | 96.34 | 98.31 | 2954 | 10.6 |
| | SSAYin et al. (2025) | 0.8 | 17.43 | 0.33 | 0.39 | 98.46 | 95.43 | 97.73 | 4193 | 12.1 |
| | SS-FGSMShi et al. (2023) | 0.35 | 15.7 | 0.16 | 0.24 | 98.66 | 95.96 | 98.08 | 2798 | 10.7 |
| | Ours | 0.82 | 27.89 | 0.87 | 1.09 | 98.1 | 92.81 | 96.8 | 1697 | 6.9 |

Table 1: Performance Comparison of Adversarial Attacks on the Brain Dataset Across Different Models.

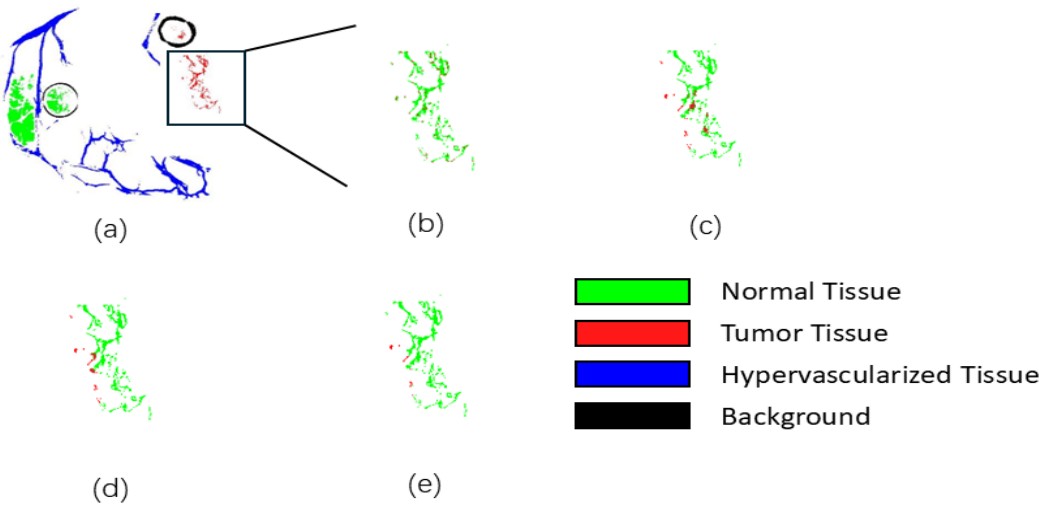

Figure 2: Classification results under different attacks and ground-truth on the brain dataset.(a)Ground-truth, (b)SS-FGSM, (c)SSA, (d)MfcaNet, (e)Ours.

**Cohen's Kappa Score** is a statistical measure of agreement between predicted and true labels, adjusted for random chance:

$$\kappa = \frac{p_o - p_e}{1 - p_e}, \tag{15}$$

where the observed agreement $p_o$ and expected agreement $p_e$ are defined as:

$$p_o = \frac{\sum_{i=1}^{C} N_{ii}}{\sum_{i=1}^{C} \sum_{j=1}^{C} N_{ij}} \tag{16}$$

| target model | attack method | Normal (↓) | Cancer (↑) | OA | AA | KAPPA | L0 | L2 |
|---|---|---|---|---|---|---|---|---|
| HybridSNRoy et al. (2019) | MfcaNetShi et al. (2022) | 0 | 84.21 | 87.11 | 57.9 | 69.47 | 1428 | 13.5 |
| | SSAYin et al. (2025) | 0 | 76.83 | 88.29 | 61.59 | 73.91 | 1764 | 13.9 |
| | SS-FGSMShi et al. (2023) | 0 | 81.54 | 87.69 | 59.23 | 71.08 | 1289 | 11.7 |
| | Ours | 0 | 89.69 | 86.55 | 55.16 | 66.47 | 752 | 7.8 |
| SSRNZhong et al. (2017) | MfcaNetShi et al. (2022) | 0 | 83.46 | 87.38 | 58.27 | 69.96 | 1387 | 12.8 |
| | SSAYin et al. (2025) | 0 | 72.72 | 89.09 | 63.64 | 76.36 | 1721 | 13.4 |
| | SS-FGSMShi et al. (2023) | 0 | 71.43 | 89.29 | 64.29 | 77.14 | 1244 | 11.2 |
| | Ours | 0 | 86.38 | 87.72 | 56.81 | 68.17 | 713 | 7.4 |
| SacNetXu et al. (2021) | MfcaNetShi et al. (2022) | 0 | 84.51 | 87.09 | 57.75 | 69.3 | 1472 | 13.9 |
| | SSAYin et al. (2025) | 0 | 80.57 | 87.72 | 59.72 | 71.65 | 1802 | 14.3 |
| | SS-FGSMShi et al. (2023) | 0 | 86.19 | 86.91 | 56.91 | 68.27 | 1326 | 12 |
| | Ours | 0 | 90.44 | 86.39 | 54.78 | 65.74 | 794 | 8.2 |
| UAGCNYu et al. (2024) | MfcaNetShi et al. (2022) | 0 | 85.12 | 87.15 | 57.82 | 69.35 | 1421 | 13.6 |
| | SSAYin et al. (2025) | 0 | 81.03 | 87.79 | 60.11 | 71.80 | 1768 | 14.2 |
| | SS-FGSMShi et al. (2023) | 0 | 86.72 | 86.98 | 56.85 | 68.40 | 1298 | 11.5 |
| | Ours | 0 | 91.23 | 86.48 | 54.92 | 66.10 | 768 | 7.9 |
| Dual-StreamYun et al. (2023) | MfcaNetShi et al. (2022) | 0 | 57.85 | 92.86 | 71.08 | 85.75 | 1398 | 13.1 |
| | SSAYin et al. (2025) | 0 | 62.13 | 92 | 68.94 | 83.25 | 1750 | 13.6 |
| | SS-FGSMShi et al. (2023) | 0 | 54.06 | 93.06 | 72.97 | 86.29 | 1279 | 11.6 |
| | Ours | 0 | 67.36 | 91.64 | 66.32 | 82.35 | 741 | 7.6 |
| RCCATu et al. (2023) | MfcaNetShi et al. (2022) | 0.4 | 47.29 | 92.68 | 71.36 | 85.03 | 1194 | 11 |
| | SSAYin et al. (2025) | 0.67 | 51.48 | 92 | 68.91 | 83.2 | 1532 | 11.2 |
| | SS-FGSMShi et al. (2023) | 0.3 | 43.63 | 93.09 | 72.69 | 86.17 | 1093 | 10.3 |
| | Ours | 1.09 | 55.32 | 91.23 | 66.34 | 81.77 | 624 | 6.9 |
| WFSSTang et al. (2024) | MfcaNetShi et al. (2022) | 0.48 | 46.98 | 92.76 | 71.51 | 85.16 | 1181 | 10.8 |
| | SSAYin et al. (2025) | 0.82 | 50.86 | 92.12 | 69.07 | 83.36 | 1519 | 11.1 |
| | SS-FGSMShi et al. (2023) | 0.32 | 42.89 | 93.17 | 73.56 | 86.62 | 1085 | 10.1 |
| | Ours | 1.14 | 54.73 | 91.33 | 66.64 | 82.11 | 611 | 6.7 |
| AIAFShi et al. (2024) | MfcaNetShi et al. (2022) | 0.5 | 25.5 | 94.88 | 86.75 | 92.18 | 1042 | 9.7 |
| | SSAYin et al. (2025) | 0.86 | 29.7 | 94.21 | 84.72 | 90.95 | 1379 | 10 |
| | SS-FGSMShi et al. (2023) | 0.34 | 21.75 | 95.25 | 88.88 | 93.17 | 987 | 9.1 |
| | Ours | 1.21 | 33.88 | 93.64 | 82.46 | 89.15 | 552 | 6.2 |
| S³ANetXu et al. (2024) | MfcaNetShi et al. (2022) | 0.58 | 24.97 | 94.93 | 87.02 | 92.39 | 1026 | 9.5 |
| | SSAYin et al. (2025) | 0.91 | 28.93 | 94.27 | 85.04 | 91.04 | 1364 | 9.8 |
| | SS-FGSMShi et al. (2023) | 0.36 | 20.94 | 95.32 | 89.53 | 93.44 | 972 | 9 |
| | Ours | 1.15 | 32.74 | 93.96 | 83.13 | 89.89 | 543 | 6.1 |

Table 2: Performance Comparison of Adversarial Attacks on the MDC Dataset Across Different Models.

$$p_e = \sum_{i=1}^{C} \left( \frac{\sum_{j=1}^{C} N_{ij}}{\sum_{i=1}^{C} \sum_{j=1}^{C} N_{ij}} \cdot \frac{\sum_{j=1}^{C} N_{ji}}{\sum_{i=1}^{C} \sum_{j=1}^{C} N_{ij}} \right) \tag{17}$$

These metrics provide a comprehensive assessment of model performance, especially under adversarial conditions where lesion misclassification must be rigorously evaluated.

### 4.4 RESULTS UNDER ATTACKS AND DEFENSES

To provide a unified and mechanism-driven analysis, we jointly examine the results presented in Table 1 (Brain dataset) and Table 2 (MDC dataset). These two tables cover multiple classifiers (HybridSN, SSRN, SACNet, Dual-Stream) and representative defense models (RCCA, WFSS, AIAF, S³ANet), enabling consistent observations across datasets and architectures. Discussing them together avoids fragmented reporting and highlights cross-cutting patterns that are obscured when each table is considered in isolation.

A central finding is a lesion-first degradation pattern: our attack drastically reduces accuracy for lesion-related classes (tumor or cancer), while global metrics (OA/AA/Kappa) remain high due to class imbalance and smoothing effects of defenses. For example, under RCCA and WFSS, OA/AA/Kappa stay in the 96–99% range, yet lesion accuracy drops most severely with our method (e.g., WFSS: 37.68% tumor accuracy on Brain, 54.73% cancer accuracy on MDC), indicating a selective but clinically critical shift toward false negatives.

It is worth emphasizing the extreme values observed in Tables 1 and 2, where some non-lesion categories show nearly 0% attack success across baseline classifiers. This does not indicate a failure of the attack but reflects two intrinsic properties of the setting. First, standard HSI classifiers such as HybridSN, SSRN, SACNet, and Dual-Stream already achieve near-perfect accuracy on clean images for non-lesion tissues (Normal, Hypervascularized, Background in Brain; Normal in MDC). Since our method is designed to perturb lesion regions only, the attack success rate on non-lesion classes naturally remains close to zero. In contrast, defense-oriented networks (RCCA, WFSS, AIAF, S$^3$ANet) cannot achieve strict 100% accuracy on clean images due to the robustness–accuracy trade-off, and thus exhibit a few non-zero misclassifications after attacks. These patterns highlight the selectivity of our attack rather than any ineffectiveness.

On the Brain dataset, tumor accuracy drops to single digits across all classifiers: 92.02% misclassification for HybridSN, 95.35% for SSRN, and 91.54% for SACNet. The lowest value for SSRN suggests that residual spectral–spatial coupling is particularly vulnerable to locally coherent, multiscale perturbations, amplifying boundary shifts in lesion regions.

On the MDC dataset (Normal vs. Cancer), our method maintains 100% accuracy for Normal while sharply degrading Cancer performance (e.g., 89.69%, 86.38%, 90.44%, and 67.36% misclassification across classifiers). This constitutes a targeted shift from positive to negative—precisely the most harmful clinical error mode—mirroring the Brain results and aligning with the design intent of our attack.

The attack also remains effective against defense networks and stronger medical architectures. With RCCA and WFSS, tumor accuracy on Brain falls to 31.41% and 37.68%, respectively, while on MDC, Cancer accuracy decreases to 55.32% and 54.73%. Similar trends are observed for AIAF and S$^3$ANet, where our method reduces Brain tumor accuracy to 29.66% and 27.89%, and MDC cancer accuracy to 33.88% and 32.74%. These results show that although defenses achieve near-perfect clean accuracy, their robustness margin against targeted perturbations remains limited.

These behaviors are consistent with the attack design. The local pixel dependency component averages gradients within small neighborhoods, preserving anatomical coherence and visual plausibility. The multiscale component injects perturbations across multiple resolutions and reprojects them back, jointly shifting decision boundaries without introducing conspicuous artifacts. This resolves the paradox of high OA/AA/Kappa alongside catastrophic lesion-class collapse.

Finally, qualitative evidence in Fig. 2 corroborates the quantitative findings: baseline methods produce only partial errors, whereas our approach induces extensive lesion misclassification while keeping least perturbations.

## 5 DISCUSSION

This study introduces a specialized adversarial attack framework specifically designed for medical hyperspectral imaging, addressing the unique spectral-spatial characteristics and Multiscale features inherent in medical data. Our innovative Local Pixel Dependency Attack leverages precise spatial relationships between neighboring pixels, while the Multiscale Information Attack strategically targets hierarchical spectral-spatial features. These innovations effectively exploit critical vulnerabilities in medical deep learning classifiers, significantly reducing classification accuracy for clinically relevant tumor regions on Brain and MDC datasets, outperforming existing methods such as SS-FGSM, SSA, and MfcaNet. However, our approach could be further enhanced by incorporating domain-specific priors, such as spectral similarity between tumor and surrounding tissues, to refine perturbation precision.

The clinical relevance of our method is substantial, as adversarial misclassifications of tumor regions can critically affect diagnostic accuracy, leading to potential misdiagnoses and compromised patient outcomes. By explicitly addressing vulnerabilities related to spectral-spatial dependencies and Multiscale information, this research highlights the urgent need for robust defensive strategies tailored specifically to medical HSI-based diagnostic systems. Future research will validate clinical applicability in diverse scenarios and develop targeted defenses to enhance medical imaging reliability.

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

## A APPENDIX

### A.1 ABLATION STUDY

In this section, we present an ablation study to evaluate the contribution of different components in our proposed adversarial attack framework. The study is divided into two parts: (1) analyzing the impact of spatial and spectral attention mechanisms, and (2) evaluating the contributions of the local pixel dependency attack and the Multiscale attack. The results of the ablation experiments are summarized in Table 3.

#### A.1.1 SENSITIVITY ANALYSIS OF SCALE FACTORS AND WINDOW SIZE

To investigate the impact of key hyperparameters in our adversarial attack framework, we conducted ablation experiments on (1) the set of scale factors $S$ used in the Multiscale Information Attack and (2) the window size $N$ employed in the Local Pixel Dependency Attack. These parameters control the granularity of Multiscale perturbations and the extent of local spatial averaging, respectively.

**(a) Effect of Scale Factors $S$:** We evaluated the framework using four different scale sets:

| Spatial Attention | Spectral Attention | Local Pixel | Multiscale | Norma Tissue($\uparrow$) | Tumor Tissue($\downarrow$) | Hyper vascularized($\uparrow$) | Background($\uparrow$) |
|---|---|---|---|---|---|---|---|
| $\checkmark$ | $\times$ | $\checkmark$ | $\checkmark$ | 85.94 | 30.67 | 84.24 | 89.76 |
| $\times$ | $\checkmark$ | $\checkmark$ | $\checkmark$ | 79.04 | 26.49 | 75.97 | 83.43 |
| $\checkmark$ | $\checkmark$ | $\checkmark$ | $\times$ | 95.31 | 15.48 | 91.74 | 94.98 |
| $\checkmark$ | $\checkmark$ | $\times$ | $\checkmark$ | 98.49 | 12.76 | 95.06 | 96.24 |
| $\checkmark$ | $\checkmark$ | $\checkmark$ | $\checkmark$ | 94.89 | **8.46** | 89.25 | 94.51 |

Table 3: Performance Comparison with Different Attention Mechanisms and Methods.

$$S_1 = \{1\}, \quad S_2 = \{1,2\}, \quad S_3 = \{1,2,4\}, \quad S_4 = \{1,2,4,8\}$$

As shown in Table 4, introducing more scales improves the effectiveness of the attack. However, excessive scaling may over-smooth the perturbation and slightly reduce attack sharpness. Therefore, $S = \{1,2,4\}$ is selected as the optimal configuration.

Table 4: Effect of Scale Factors $S$ on Attack Effectiveness (Tumor Class Accuracy $\downarrow$)

| Scale Factors $S$ | Tumor Acc. (%) $\downarrow$ |
|---|---|
| $\{1\}$ | 50.87 |
| $\{1, 2\}$ | 27.43 |
| $\{1, 2, 4\}$ | **9.27** |
| $\{1, 2, 4, 8\}$ | 16.36 |

**(b) Effect of Window Size $N$:**

To assess the sensitivity to local spatial context, we varied the window size $N$ in the Local Pixel Dependency Attack as:

$$N = 3 \times 3, \ 5 \times 5, \ 7 \times 7, \ 11 \times 11$$

As shown in Table 5, moderate window sizes such as $5 \times 5$ offer the best trade-off between spatial coherence and attack precision. Larger windows may dilute local structures, weakening the perturbation's targeting power.

Table 5: Effect of Window Size $N$ on Attack Effectiveness (Tumor Class Accuracy $\downarrow$)

| Window Size $N$ | Tumor Acc. (%) $\downarrow$ |
|---|---|
| $3 \times 3$ | 28.74 |
| $5 \times 5$ | **12.55** |
| $7 \times 7$ | 19.49 |
| $11 \times 11$ | 26.68 |

### A.1.2 ABLATION STUDY ON ATTENTION MECHANISMS AND ATTACK COMPONENTS

We systematically evaluate the contributions of spatial and spectral attention mechanisms and core attack components through ablation studies in Table 3. When removing both spatial and spectral attention, tumor classification accuracy rises to 30.67%, indicating degraded feature detection capability. Isolating spectral attention removal further degrades performance (26.49% tumor accuracy), underscoring its critical role in leveraging spectral dependencies. Incorporating both mechanisms significantly enhances attack effectiveness, reducing tumor accuracy to 12.76%.

For core attack components, the Multiscale attack alone achieves 15.48% tumor accuracy by disrupting multiresolution features, while the local pixel attack reaches 12.76% by exploiting spatial dependencies. Their synergistic combination maximizes impact, reducing tumor accuracy to 8.46%, a 44.5% improvement over individual components. These findings quantitatively validate the complementary roles of attention mechanisms and attack strategies in exploiting MHSI vulnerabilities.

### A.1.3 OVERALL EVALUATION

Combining spatial/spectral attention and local - pixel/Multiscale attack strategies yields the most potent adversarial attack. This setup hits the lowest tumor classification accuracy (8.46%) while strongly degrading accuracy across other classes, highlighting the need to integrate these components to fully exploit model vulnerabilities.

Our ablation study shows each framework component boosts effectiveness, with maximal impact when used together. These insights stress the superiority of our integrated approach in exposing MHSI classifier vulnerabilities, underscoring the need for robust, tailored defenses.

