# OpenReview forum: "Adversarial Attacks on Medical Hyperspectral Imaging Exploiting Spectral-Spatial Dependencies and Multiscale Features"
_ICLR.cc/2026/Conference — ICLR 2026 Conference Desk Rejected Submission_

### Official Review · Reviewer_KRRQ · 2025-10-28

**Soundness:** 2
**Presentation:** 2
**Contribution:** 2
**Rating:** 4
**Confidence:** 3

**Summary:**

The paper targets adversarial robustness in medical hyperspectral imaging (MHSI) classification. It introduces two training-free attacks: Local Pixel Dependency Attack (LPDA), which averages gradients in local windows to preserve spatial coherence, and Multiscale Information Attack (MIA), which injects perturbations across down-/up-sampled scales and aggregates them. Experiments on the Brain and MDC datasets, with several classifiers and defense networks, show that these attacks sharply degrade tumor-class accuracy while keeping global OA/AA/Kappa nearly unchanged. Appendix studies window and scale sensitivity and component ablation.

**Strengths:**

Addresses a practically significant yet under-explored topic: robustness of MHSI models.

Exploits spatial and multiscale dependencies that align with medical-image classification characteristics. Addresses a practically significant yet under-explored topic: robustness of MHSI models.

**Weaknesses:**

Lacks quantitative and visual assessment of perturbation imperceptibility.

No adaptive-attack evaluation against defenses; robustness conclusions remain incomplete.

Several symbols and implementation details are ambiguous (local-gradient computation, scale-set definition).

Methodological innovation and theoretical explanation are limited.

**Questions:**

Provide quantitative metrics (PSNR/SSIM/spectral-curve shift) and ROI visualizations of perturbation imperceptibility.

Describe and evaluate adaptive attacks tailored to RCCA, WFSS, AIAF, and S3ANet.

Specify the position-dependent gradient computation and parameter settings (window, stride, iterations).

Define the scale set S and interpolation operators precisely.

Add mechanism-level evidence (feature-response or gradient heatmaps, cross-scale disruption visualization).

---

### Official Review · Reviewer_12ny · 2025-10-31

**Soundness:** 3
**Presentation:** 3
**Contribution:** 3
**Rating:** 4
**Confidence:** 3

**Summary:**

The paper explores the adversarial threat against deep learning-based Medical Hyperspectral Imaging (HSI). Specifically, the paper studies two fundamental reasons for the medical HSI models' vulnerability: dependence on local pixel dependencies and reliance on multiscale spectral-spatial features. To this end, the paper introduces an adversarial attack scheme specifically against medical HSI by exploiting the spatial connection between neighborhood pixels associated with multiscale information. Thus, the attack perturbs both spectral and spatial features across hierarchical scales. Extensive experiments across the brain and the multi-dimensional choledoch database have demonstrated the efficacy of the proposed medical adversarial attack scheme. Furthermore, systematic analyses have also justificated the effectiveness of the method design.

**Strengths:**

1. The paper is generally well-written. The motivation and the medical attack topic are interesting. It's also significant to highlight the adversarial security threat of the modern deep learning-based diagnosis system.
2. The experimental results are promising. The proposed method achieves superior attack efficacy across diverse benchmarks and scenarios.
3. Detailed discussion of the importance of adversarial threats in the context of computer-aided medical image analysis is concrete and insightful.
4. The experimental setup and evaluation metric introduction are comprehensive.

**Weaknesses:**

1. The proposed method seems to focus solely on the white-box scenarios. However, most of the deep diagnosis systems/models are in a black box in practice. Thus, the paper should additionally discuss the black-box extension.
2. The paper primarily focuses on the empirical justification of the spatial-spectral features, yet the theoretical analyses are limited. This somehow restricted the efficacy justification of the proposed method.
3. Visualizations of adversarial attacks for medical images are missing. The paper primarily focuses on the quantitative results. More insights from the qualitative results should be given.
4. It seems that the proposed method is also applicable to natural images, yet few discussions about the specificity of the proposed method in the medical image analysis area are included.

**Questions:**

1. Can the proposed method be extended to a black-box scenario for a real-world attack simulation?
2. Is it possible to include error bar tests to evaluate the stability of the proposed adversarial attack method?
3. Can the proposed method be extended to the natural image domain to conduct adversarial attacks? If so, then what's the unique contribution to the medical domain?

---

### Official Review · Reviewer_L9wW · 2025-11-02

**Soundness:** 2
**Presentation:** 2
**Contribution:** 2
**Rating:** 2
**Confidence:** 4

**Summary:**

This paper investigates the vulnerabilities of deep learning models used in medical hyperspectral imaging (HSI), revealing that their reliance on local pixel dependencies and multiscale spectral-spatial features makes them highly susceptible to adversarial attacks. The authors introduce two novel attack methods—Local Pixel Dependency Attack and Multiscale Information Attack—which exploit spatial relationships and hierarchical features to generate imperceptible perturbations that drastically reduce classification accuracy. Experiments on brain and cancer datasets demonstrate that these attacks outperform existing methods.

**Strengths:**

+) The paper is well-organized and easy to follow

+) An important topic and raising awareness of the risks posed by adversarial attacks in sensitive medical applications

**Weaknesses:**

-) The discussion is limited in the digital domain, and may not be feasible and practical to perform these attacks in real-world hospital environments

-) The proposed attacks focus primarily on classification tasks; their impact on other medical imaging tasks (e.g., segmentation, detection) is not explored.

**Questions:**

a) What defense mechanisms can we perform to mitigate these specific attacks in clinical settings?

b) How feasible is it for an attacker to deploy these methods in real-world hospital environments?

c) Do the proposed adversarial attacks perform on other medical imaging tasks, such as segmentation or detection?

---

### Official Review · Reviewer_WyoT · 2025-11-06

**Soundness:** 2
**Presentation:** 2
**Contribution:** 2
**Rating:** 4
**Confidence:** 3

**Summary:**

The paper introduces a novel adversarial attack framework for medical hyperspectral imaging (MHSI), focusing on vulnerabilities arising from local pixel dependencies and multiscale spectral-spatial features. It proposes two attack methods: the Local Pixel Dependency Attack, which exploits spatial relationships between neighboring pixels, and the Multiscale Information Attack, which introduces perturbations across multiple resolutions to target hierarchical spectral and spatial features. The authors demonstrate the effectiveness of these attacks through experiments on the Brain and MDC datasets, showing that their methods significantly reduce classification accuracy in clinically relevant areas, such as tumor detection, while keeping perturbations imperceptible. The paper also includes an ablation study, confirming the enhanced effectiveness of the combined attack strategies. While the paper addresses important issues in MHSI, the methods proposed are not sufficiently novel, as they largely build on existing adversarial attack techniques without offering a breakthrough specific to medical hyperspectral imaging.

**Strengths:**

1. This paper focus on an interesting research area.
2. The motivation of this work is clear and convince.

**Weaknesses:**

1. While the paper introduces the Local Pixel Dependency Attack and Multiscale Information Attack, these techniques are not significantly novel in the broader context of adversarial attacks. Similar methods, such as Spectral-Spatial FGSM (SS-FGSM) and Spectral-Spatial Attack (SSA), already explore pixel-level and spectral-spatial perturbations in hyperspectral imagery.
2. The proposed attack methods, such as the Local Pixel Dependency Attack and Multiscale Information Attack, rely heavily on gradient-based perturbations, which are commonly used in many adversarial attack frameworks.
3. While the Local Pixel Dependency Attack exploits local pixel relationships, the method could benefit from a more sophisticated model of spatial dependencies. The attack assumes that the spatial context can be effectively modeled with a simple local window (e.g., 3x3, 5x5). However, in medical hyperspectral images, spatial dependencies may be more complex and non-linear, especially in tissue boundaries, tumor regions, or vascular networks.
4. The current framework focuses on attacking a single model at a time, but adversarial attacks are often evaluated for their transferability across different models. In real-world scenarios, medical hyperspectral imaging models may vary in architecture and training methodology, making it crucial to assess whether the proposed attacks are model-agnostic.

**Questions:**

1. Have the authors considered testing the transferability of the attacks across different model architectures (e.g., CNNs, transformers, etc.)? If the attacks work primarily on one model but fail on others, how generalizable do the authors believe their framework is for real-world clinical systems?
2. Did the authors consider prioritizing specific spectral bands that might have higher diagnostic value, particularly in medical applications like tumor detection?
3. How do the authors balance the attack's strength with the need for imperceptibility, especially in clinical settings where the perturbations must be indistinguishable from real images?

---

### Note · Program_Chairs · 2026-01-17
**Submission Desk Rejected by Program Chairs**

The following references in this submission do not refer to real documents and/or have major errors in bibliographic information:

 Zhaoxia Yin, Lichun Tang, Cong Kong, Hang Su, and Bin Luo. Sparse adversarial attack method for deep learning hyperspectral image classification models. CN Patent: CN117079137B, 2025. Granted patent.